# Evaluation of the Safety and Gastrointestinal Migration of Guanidinylated Chitosan after Oral Administration to Rats

**DOI:** 10.3390/jfb14070340

**Published:** 2023-06-27

**Authors:** Nowshin Farzana Khan, Hideaki Nakamura, Hironori Izawa, Shinsuke Ifuku, Daisuke Kadowaki, Masaki Otagiri, Makoto Anraku

**Affiliations:** 1Faculty of Pharmaceutical Sciences, Sojo University, 4-22-1 Ikeda, Nishi-ku, Kumamoto 860-0082, Japan; 2Faculty of Engineering, University of Miyazaki, 1-1 Gakuen Kibanadai-Nishi, Miyazaki 889-2192, Japan; 3Graduate School of Engineering, Tottori University, 4-101 Koyama-cho Minami, Tottori 680-8552, Japan

**Keywords:** chitosan, guanidinylation, membrane permeability, water-soluble

## Abstract

Arginine-rich membrane-permeable peptides (APPs) can be delivered to cells by forming complexes with various membrane-impermeable bioactive molecules such as proteins. We recently reported on the preparation of guanidinylated chitosan (GCS) that mimics arginine peptides, using chitosan, a naturally occurring cationic polysaccharide, and confirmed that it enhances protein permeability in an in vitro cell system. However, studies on the in vivo safety of GCS are not available. To address this, we evaluated the in vivo safety of GCS and its translocation into the gastrointestinal tract in rats after a single oral administration of an excessive dose (500 mg/kg) and observed changes in body weight, major organ weights, and organ tissue sections for periods of up to 2 weeks. The results indicated that GCS causes no deleterious effects. The results of an oral administration of rhodamine-labeled chitosan and an evaluation of its migration in the gastrointestinal tract suggested that the disappearance of rhodamine-labeled GCS from the body appeared to be slower than that of the non-dose group and pre-guanidinylated chitosan due to its mucoadhesive properties. In the future, we plan to investigate the use of GCS to improve absorption using Class III and IV drugs, which are poorly water-soluble as well as poorly membrane-permeable.

## 1. Introduction

Chitosan, a partially deacetylated product derived from chitin, is composed of glucosamine and *N*-acetyl-glucosamine units and is a biocompatible, biodegradable, safe, and low-toxicity polymer that is used in a variety of applications [1,2,3,4]. Due to its low immunogenicity, especially for drug administration via the oral route, this polymer is widely used as a carrier for low-molecular-weight drugs, vaccines, and DNA in the medical and pharmaceutical industries [1,5,6]. It is also being increasingly used in the food, cosmetic, and agricultural fields. In addition to being listed on the Ministry of Health and Welfare’s list of food additives, it is also expected to be applied to food for specified health uses and intestinal regulators by taking advantage of its characteristics, such as its status as one of the few naturally occurring basic amino-polysaccharides in nature [7,8,9].

Low-molecular-weight chitosan is an attractive material with the potential for improving the absorption of poorly soluble drugs and proteins/peptides from the gastrointestinal tract because it can increase their membrane permeability [10,11,12,13]. Similar to arginine peptides, guanidinylation of the amino groups of chitosan would also likely enhance the permeability of drugs and peptides through biological membranes. Given these predictions, we recently developed a simple method for the guanidinylation of CS with 1-amidinopyrazole hydrochloride (AP) that does not require complex procedures [14,15]. The in vitro safety of GCS, including its stability at various pH levels, has already been confirmed [16]. Furthermore, the lower acidity of the guanidino group (pKa: 12.5 as protonated form) compared to the amino group (pKa: 6.5 as protonated form) of GCS is also likely to enhance electrostatic interactions with protein/peptide drugs in biological conditions. In fact, it has been reported that arginine peptides with guanidino groups are attracted to the cell surface by interaction with cell surface proteoglycans and are then translocated into the cell via two different pathways, endocytosis, and direct plasma membrane permeation [17,18,19,20]. The GCS mimicking arginine peptide prepared by this method not only binds more efficiently to albumin than chitosan before guanidinylation but was also found to enhance the biological membrane permeability of albumin by GCS itself and by GCS in uptake experiments using Hella cells [16]. We, therefore, conclude that the use of GCS as an oral administration aid has considerable potential. On the other hand, considering the actual oral administration, there are concerns regarding the toxicity of GCS itself and its effects on the body. However, only a few in vivo studies of GCS have been reported.

In general, no toxicity has been reported for chitosan, even at high doses (500 mg/kg), in experiments with mice, rats, and dogs. Therefore, in the present study, an excessive oral single dose (500 mg/kg) was administered to rats and, based on this dosage, its safety in the rat body was compared with that of chitosan before guanidinylation in terms of changes in serum parameters and changes in weight and the morphology of major organs. Chitosan labeled with rhodamine was also administered orally, and its migration in the gastrointestinal tract was compared with that of chitosan before guanidinylation using CS lactate as the marker.

## 2. Materials and Methods

### 2.1. Materials

CS lactate (molecular weight about 7600) was purchased from Koyo Kagaku K.K. (Tottori, Japan) and contained a 31.2% fraction (elemental analysis) that was not deacetylated [16]; AP was purchased from Tokyo Kasei Kogyo Co. Rhodamine was purchased from Sigma-Aldrich (St. Louis, MO, USA). All other reagents were of commercial grade.

### 2.2. Preparation and Characterization of GCS

GCS was performed according to the method of Izawa et al. [16]. Specifically, CS lactate (1.00 g) was dissolved in distilled water; AP (1.75 g, 11.9 mmol) was then added to the CS solution and the mixture was stirred for 10 min. Triethylamine (1.45 g, 16.3 mmol) was added and the mixture was stirred vigorously (800 rpm) at room temperature for 7 days. The reaction mixture was then poured into a large volume of 2-propanol, and the resulting precipitate was collected by centrifugation and washed with methanol. The product was purified by dialysis through a Bisking tube (Mw cutoff: 3500) in a large volume of water. The yield was 72.0%. The degree of guanidinylation, as estimated from the elemental analysis, was 48.2 mol%. From the solubility and stability tests of various ratios of GCS, GCS in this ratio was adopted [16]. The zeta potentials of the purified GCS and CS lactates were measured using a Zeta sizer (Nano-zs, Sysmex Co., Ltd., Tokyo, Japan) after dissolving the samples in acetic acid buffer (pH 4.5).

### 2.3. Animals

Animals were 5-week-old SD rats (*n* = 12) purchased from the Japan SLC Corporation (Shizuoka, Japan) and kept acclimated with basic feed (CE-2 solid feed, Tokyo, CLEA Japan, Inc, Japan) and tap water for 6 days. All animals were weighed and divided into 3 groups (4 males per group) to avoid weight bias among the groups. The animals were housed in a rearing room with a pariah system under the following conditions: room temperature of 24 ± 1 °C, humidity of 55%, ventilation frequency of 18 cycles (all fresh), 12 h of fluorescent lighting, and 12 h of lights out. Animals were housed in transparent polycarbonate cages (26 cm wide, 42 cm long, and 17 cm high), four by four, and the bedding was soft chips from Sankyo Lab Service Co. As drinking water, tap water was administered ad libitum throughout the study period. Animal experiments were conducted in accordance with the general principles of scientific research and under the Ethics Committee for Laboratory Animals of Sojo University (2022-p-001), keeping in mind the consideration to protect animal welfare based on the basic perspective of respecting animal life.

### 2.4. Safety Studies in Rats

Each of the three groups of males was treated with the test substance. Two groups were given a single dose of 500 mg/kg each of CS lactate and GCS as a water dispersion, the maximum dose utilized in the natural chitosan subacute study. For the remaining group, only water was administered as a single dose as a control. The general condition of all animals was observed daily during the study period, and their body weights were measured daily. Two weeks after administration, the abdomen was opened under isoflurane anesthesia, blood was collected from the abdominal aorta, and the animals were bled, sacrificed, and divided. Each organ was observed endoscopically to confirm that there were no major changes. The main organs (liver, kidney, and spleen) were removed and fixed in 10 % neutral buffered formalin solution. These organs were then excised and thin sections were prepared by the usual method and stained with hematoxylin and eosin (H.E.) for histopathological examination. Blood samples were obtained from Fujifilm Hematology and assayed for various biochemical parameters.

### 2.5. Evaluation of the Behavior of Rhodamine-Labeled GCS and CS Lactate in the Gastrointestinal Tract in Rats

Rhodamine-labeled chitosan (10 mg/kg) was orally administered to DDY mice (10 weeks old, male) for each chitosan sample [16], and the movement of the sample in the digestive tract 10 min after administration was determined using IVIS Lumina XR (Perkin Elmer Japan Inc., Yokohama, Japan) (3 males per group). The average fluorescence intensity was measured from the stomach side of the intestinal fluorescence image taken with IVIS. The ratio of fluorescence intensity at each location to the total fluorescence intensity was calculated, and the fluorescence intensity fractions were plotted as a function of distance from the stomach; fluorescence intensities in the 5–15 cm or 17.5 cm–30 cm interval were integrated and marked.

### 2.6. Statistical Analysis

All experimental data are presented as the mean ± standard deviation or standard error. The one-way ANOVA with Tukey–Kramer method was used to test for significant differences. A statistically significant difference was evaluated when the risk value was less than 0.05.

## 3. Results

### 3.1. Preparation and Characterization of GCS

The same preparation method was used to prepare this product, and the NMR and IR spectra confirm that a similar GCS had been formed [16]. The ζ potential of GCS was markedly positive compared to lactic CS (Table 1, *p* < 0.01). Normally, the intracellular potential is kept lower than the extracellular one, and GCS, which is more polycationic in nature than lactated CS, would be expected not only to adsorb more strongly to the biomembrane through electrostatic interactions but also to be rapidly and more efficiently taken up in the direction of lower potential, i.e., inside the cell, driven by the potential difference. Thus, GCSs with a high zeta potential have the potential for functioning as membrane-permeable peptides.

### 3.2. Changes in Body Weight, Hematological Parameters, and Major Organs at 2 Weeks after the Administration of GCS and CS Lactate

No significant changes were observed in the general condition of the animals in any of the groups during the study period, and all animals survived until the end of the study. The changes in body weights of each group during the study period are shown in Figure 1A. No statistically significant differences were observed between the control and test substance-treated groups. The results of hematological and serum biochemical tests are shown in Table 2. The triglycerides were reduced in the GCS-treated group compared with the untreated group. Weight changes in the major organs such as the liver, spleen, and kidney are shown in Figure 1B. The results showed no difference in the weight of major organs compared to the control group. Although not shown in the data, no differences were observed visually in the other organs compared to the control group.

Histopathological observations showed that in the major tissues of the liver, spleen, and kidneys of all groups of animals, including the control group, no cellular or nuclear disorganization or cytoplasmic hollowing was found in the liver, nor was glomerular atrophy or tubular dilation found in the kidney (Figure 2). These results indicate that there was no difference between the control group and the group that had been treated with the test substance.

### 3.3. Gastrointestinal Retention of Rhodamine-Labeled Chitosan

The retention of rhodamine-labeled GCS and CS lactate in the gastrointestinal tract in rats was compared (Figure 3). The findings indicate that rhodamine-labeled GCS remained in the upper part of the small intestine at 10 min after administration, while rhodamine-labeled CS lactate moved to the lower part of the small intestine (Figure 3C), confirming that the rhodamine-labeled GCS is more efficiently retained in the digestive tract compared to that of rhodamine-CS lactate.

## 4. Discussion

In recent years, membrane-permeable peptides, collectively referred to as cell-penetrating peptides (CPPs), have often been used to introduce bioactive substances into cells [17,18,19,20]. This method allows for the efficient delivery of high-molecular-weight drugs and nanoparticles, which are normally unable to penetrate the cell membrane, into cells by covalent (chemical bonding) cross-linking or by forming stable non-covalent complexes with CPPs. Fusion proteins with CPPs can also be used for the intracellular delivery of proteins. The number of applications of CPPs, including arginine peptides, has been increasing year by year due to their ease of use and the ease of obtaining results at the cellular level [21,22].

Izawa et al. also recently prepared a membrane-permeable peptide-mimicking GCS using lactated chitosan; GCS showed 6-fold higher translocation into HeLa cells compared to CS without guanidine groups [16]. This clearly indicates that guanidinylation enhances cellular uptake. We also found that GCS, when simply added to a transport medium (pH 7.4) containing bovine serum albumin (BSA) significantly promoted the internalization of BSA, and the amount of internalization in the presence of WGCS was 5-fold higher than in the presence of CS. This may be due to the more efficient binding of WGCS and BSA due to the electrostatic interactions of the guanidino groups [16]. These results indicate that GCS, like CPP, may function as an aid to improve drug permeability and absorption. On the other hand, although the synthesis of GCS using AP is simple, there might be concerns regarding the toxicity of the reagent. Therefore, we conducted a safety study on rats based on a maximum dose of 500 mg/kg, a concentration that has also been used in subacute toxicity studies of natural chitosan.

In the present study, a 2-week acute toxicity study was conducted in SD rats using a single oral dose (500 mg/kg) of GCS. The results showed that there were no obvious differences in the general condition and food intake of the rats as a result of the administration of the test substance. Serum biochemical tests showed that triglyceride levels were decreased in the GCS-treated group. In addition, liver parameters such as AST, ALT, and ALP levels showed a decreasing trend in the GCS-treated group compared to the untreated group (Table 1). Although only a single dose of GCS and CS lactate was administered in this study, the large dose resulted in a long residence time in the gastrointestinal tract, suggesting that the GCS group may have inhibited the absorption of some of the food due to the long residence time in the gastrointestinal tract, although the food intake did not change among the rats. In fact, when rhodamine-labeled GCS was orally administered to rats, gastrointestinal tract retention was significantly prolonged compared to rhodamine-labeled CS lactate (Figure 3). In addition, a comparison of zeta potentials between GCS and lactated CS at this time showed that GCS has a remarkable positive charge. These results suggest that the decrease in blood triglyceride levels in the GCS-treated group that was observed in this experiment may be due to the protective effect of GCS on the mucosal surface in the digestive tract by delaying the flowability and increasing the adhesive property of GCS in the digestive tract.

These results indicate that GCS can be safely administered to rats despite the administration of excess GCS used in the present study suggesting that GCS may be used to improve the absorption of not only peptides and other protein drugs but also other water- and membrane-insoluble drugs.

## 5. Conclusions

GCS, which mimics an arginine peptide, was prepared using chitosan, a natural cationic polysaccharide, and its in vivo safety and gastrointestinal transfer were investigated in rats. The results showed that the safety of GCS was the same as that of CS lactate without and before guanidinylation, and the safety of GCS was confirmed. Since the ratio of drug to membrane-permeable peptide is usually around 1:1 to 1:5, it would be expected to be, at most, one-tenth of the present dose. Therefore, ensuring safety at the present dosage level will greatly contribute to the promotion of future research. In the future, we plan to investigate the effect of the membrane permeability of GCS on the absorption of peptides and water-soluble drugs from the gastrointestinal tract.

## Figures and Tables

**Figure 1 jfb-14-00340-f001:**
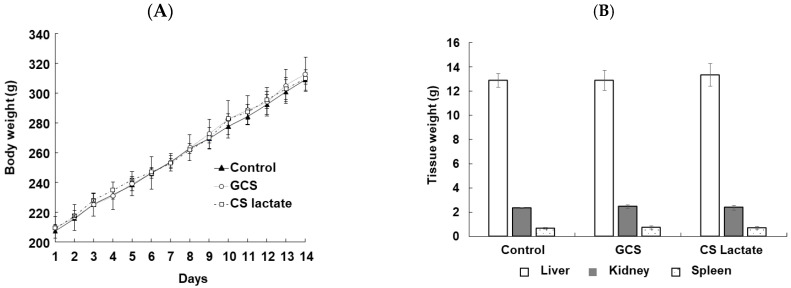
Determination of changes in the body (**A**) and major tissue (**B**) weight of rats following the single oral administration (500 mg/kg) of GCS and CS lactate. Values are expressed as mean ± SD (*n* = 4). *p*-values were determined using one-way ANOVA.

**Figure 2 jfb-14-00340-f002:**
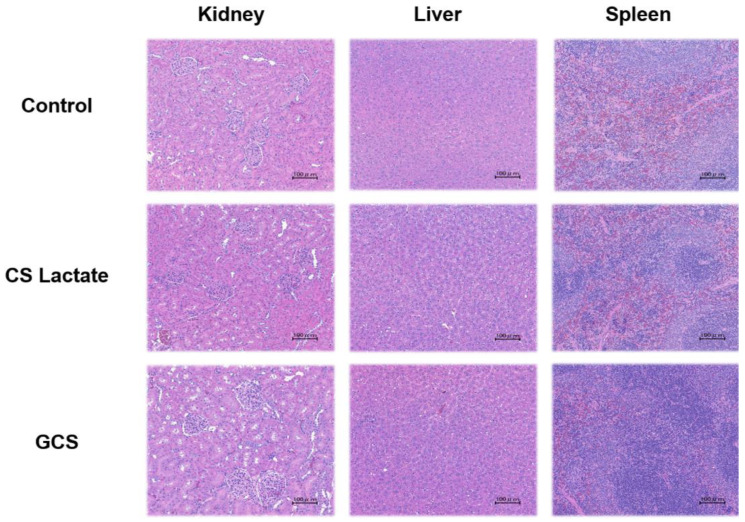
Histopathological changes in kidney, liver, and spleen tissues in SD rats with or without GCS and CS lactate after 2 weeks. HE staining of each tissue (scale bars indicate 100 μm).

**Figure 3 jfb-14-00340-f003:**
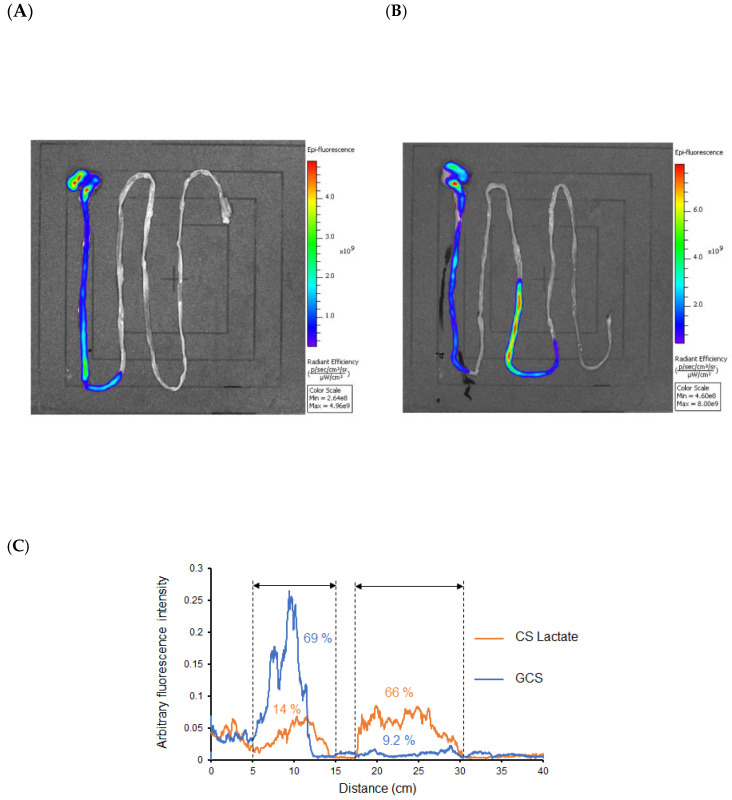
Gastrointestinal migration of rhodamine-labeled GCS (**A**) and CS lactate (**B**) after oral administration (10 min). The fluorescence intensity distribution of GCS and CS in the stomach to the intestine (**C**) (*n* = 3).

**Table 1 jfb-14-00340-t001:** Zeta potential of CS Lactate and GCS.

Formulations	Zeta Potential (mv)
CS Lactate	13.29 ± 1.96
GCS	19.14 ± 0.86 *

Values are expressed as mean ± SD (*n* = 4). *p*-values were determined using one-way ANOVA. For all analyses, * *p*  <  0.01 was considered to indicate statistical significance.

**Table 2 jfb-14-00340-t002:** Blood parameters change in rats following the single oral administration (500 mg/kg) of GCS and CS lactate.

Blood Parameters	Control	GCS	CS Lactate
Total Protein (g/dL)	5.4 ± 0.3	5.4 ± 0.3	5.4 ± 0.2
Albumin (g/dL)	4.0 ± 0.2	3.8 ± 0.2	3.9 ± 0.1
AST(GOT) (U/L)	76 ± 22	57 ± 6.7	68 ± 13
ALT(GPT) (U/L)	50 ± 13	42 ± 2.2	45 ± 5.4
LDH (U/L)	112 ± 16	87 ± 27	107 ± 44
ALP (U/L)	446 ± 122	371 ± 43	400 ± 88
Amylase (U/L)	3000 ± 613	2739 ± 179	2936 ± 491
Lipase (U/L)	9.0 ± 0.8	8.5 ± 0.6	8.0 ± 0.8
Urea Nitrogen (mg/dL)	19 ± 2.1	19 ± 0.4	19 ± 0.7
Creatinine (mg/dL)	0.2 ± 0.01	0.2 ± 0.03	0.2 ± 0.03
Total Cholesterol (mg/dL)	65 ± 6.2	66 ± 7.8	55 ± 17
triglycerides (mg/dL)	101 ± 21	63 ± 20	114 ± 57
Sodium (mEq/L)	142 ± 2.2	141 ± 0.8	143 ± 1.9
Potassium (mEq/L)	5.8 ± 2.6	5.8 ± 2.0	6.4 ± 1.8
Chlorine (mEq/L)	101 ± 1.7	101 ± 1.3	102 ± 1.2
Calcium (mg/dL)	10.4 ± 0.5	10.3 ± 0.3	10.5 ± 0.1
Inorg. Phosphorus (mg/dL)	8.4 ± 1.6	9.0 ± 1.4	8.9 ± 0.8
Blood Sugar (mg/dL)	210 ± 25	201 ± 13	210 ± 29
Total bile acid (μmol/L)	15.0 ± 13	13.6 ± 11	14.1 ± 01
Total Bilirubin (mg/dL)	<0.1	<0.1	<0.1

Values are expressed as mean ± SD (*n* = 4). *p*-values were determined using one-way ANOVA.

## Data Availability

The datasets generated and/or analyzed during the current study are available from the corresponding author upon reasonable request.

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
