# Peer review of "Evaluation of the Safety and Gastrointestinal Migration of Guanidinylated Chitosan after Oral Administration to Rats"

_jfb, 2023, doi:10.3390/jfb14070340_

Round 1

Reviewer 1 Report

In the submitted manuscript “Evaluation of the safety and gastrointestinal migration of guanidinylated chitosan after oral administration to rats”, the authors have assessed the safety profile of the guanidinylated chitosan by oral administration.

The manuscript is of interest but requires some revisions before publications.

11.   The authors assert GCS being a potential oral delivery vehicle but does not provide its own GI stability in harsh GI conditions. Provide information of any studies conducted on stability.

22.  Authors should provide more information why GCS was selected. Possibly provide more examples of studies where GCS have been studied as a vehicle in in-vivo studies.

33.     Provide the yield of GCS in % terms as well. Also provide what is the desired or optimum degree of guanidinlyation. Have the authors conducted some in-vitro studies comparing the effects of difference on degree of guanidinlyation on delivery, membrane permeability?

44. Authors should explain the relevance of zeta potential of GCS in results section. Also, the sentences GCS had been formed needs to be restructured.

55.  Author should provide some insights into decrease levels of TG with GCS administration, what might be the mechanism of it. Also, do the author expects similar decrease of TG levels on lower dose of GCS.

66.   Also, for oral administration of GCS as a vehicle with peptides/protein drug, what would be the optimum dose? At a lower dose of GCS as delivery agent, would it have same GIT retention property?

17.  Provide a quantitative data for fluorescence of GCS-rhodamine and CS in GIT to show different retention times.

Overall the manuscript is well written but in some places the language can be improved. 

Author Response

Reply to comments from reviewers:

We are appreciative of the reviewer’s helpful comments regarding our manuscript“Evaluation of the safety and gastrointestinal migration of guanidinylated chitosan after oral administration to rats” (Manuscript Number: jfb-2458289). We carefully considered their comments and prepared a revised version of our manuscript that takes these comments into account. The following is a point-by-point list of responses to the comments. In the revised manuscript, the revised sections or sentences are highlighted in red.

Reply to Reviewer #1:

We much appreciate the reviewers for these positive and constructive comments.

  • In response to the comment, “The authors assert GCS being a potential oral delivery vehicle but does not provide its own GI stability in harsh GI conditions. Provide information of any studies conducted on stability.”, we already published GCS’s GI stability by ref. 16. To address this, we added some additional text in Introduction as follows:

“The in vitro safety of GCS, including its stability at various pH levels, has already been confirmed [16].”

  • In response to the comment, “Authors should provide more information why GCS was selected. Possibly provide more examples of studies where GCS have been studied as a vehicle in in-vivo studies.”, we added some additional text in the Introduction as follows:

“In fact, it has been reported that arginine peptides with guanidino groups are attracted to the cell surface by interaction with cell surface proteoglycans, and are then translocated into the cell via two different pathways, endocytosis, and direct plasma membrane permeation [17-20]. The GCS mimicking arginine peptide prepared by this method not only binds more efficiently to albumin than chitosan before guanidinylation but was also found to enhance the biological membrane permeability of albumin by GCS itself and by GCS in uptake experiments using Hella cells [16]. We, therefore, conclude that the use of GCS as an oral administration aid has considerable potential. On the other hand, considering the actual oral administration, there are concerns regarding the toxicity of GCS itself and its effects on the body. But, only a few in vivo studies of GCS have been reported.”

  • In response to the comment, “Provide the yield of GCS in % terms as well. Also provide what is the desired or optimum degree of guanidinlyation. Have the authors conducted some in-vitro studies comparing the effects of difference on degree of guanidinlyation on delivery, membrane permeability?”, we added some additional text in the Materials and Methods as follows:

“The yield was 72.0%. The degree of guanidinylation, as estimated from the elemental analysis, was 48.2%. From the solubility and stability tests of various ratios of GCS, GCS in this ratio was adopted [16].”.

  • In response to the comment, “Authors should explain the relevance of zeta potential of GCS in results section. Also, the sentences GCS had been formed needs to be restructured.”, we added some additional text in the results as follows:

“Normally, the intracellular potential is kept lower than the extracellular one, and GCS, which is more polycationic in nature than lactated CS, would be expected not only to adsorb more strongly to the biomembrane through electrostatic interactions but also to be rapidly and more efficiently taken up in the direction of lower potential, i.e., inside the cell, driven by the potential difference. Thus, GCSs with a high zeta potential have the potential for functioning as membrane-permeable peptides.”

  • In response to the comment, “Author should provide some insights into decrease levels of TG with GCS administration, what might be the mechanism of it. Also, do the author expects similar decrease of TG levels on lower dose of GCS.”, we added some additional text in the results and discussion as follows:

“Normally, the intracellular potential is kept lower than the extracellular one, and GCS, which is more polycationic in nature than lactated CS, would be expected not only to adsorb more strongly to the biomembrane through electrostatic interactions but also to be rapidly and more efficiently taken up in the direction of lower potential, i.e., inside the cell, driven by the potential difference. Thus, GCSs with a high zeta potential have the potential for functioning as membrane-permeable peptides.”

“In fact, when rhodamine-labeled GCS was orally administered to rats, the gastrointestinal tract retention was significantly prolonged compared to rhodamine- CS lactate (Fig.3). In addition, a comparison of zeta potentials between GCS and lactated CS at this time showed that GCS has a remarkable positive charge. These results suggest that the decrease in blood triglyceride levels in the GCS-treated group that was observed in this experiment may be due to the protective effect of GCS on the mucosal surface in the digestive tract by delaying the flowability and increasing the adhesive property of GCS in the digestive tract.”

  • In response to the comment, “Also, for oral administration of GCS as a vehicle with peptides/protein drug, what would be the optimum dose? At a lower dose of GCS as delivery agent, would it have same GIT retention property?”, we added these sentences in conclusions as follows:

“Since the ratio of drug to membrane-permeable peptide is usually around 1:1 to 1:5, it is expected to be at most one-tenth of the present dose. Therefore, ensuring safety at the present dosage level will greatly contribute to the promotion of future research.”

  • In response to the comment, “Provide a quantitative data for fluorescence of GCS-rhodamine and CS in GIT to show different retention times.”, we added the Methods and results in Fig.3C and as follows:

“The average fluorescence intensity was measured from the stomach side of the intestinal fluorescence image taken with IVIS. The ratio of fluorescence intensity at each location to the total fluorescence intensity was calculated, and the fluorescence intensity fractions were plotted as a function of distance from the stomach; fluorescence intensities in the 5-15 cm or 17.5 cm-30 cm interval were integrated and are marked.”

“The fluorescence intensity distribution of GCS and CS in the stomach to the intestine (C) (n=3)”.

Reviewer 2 Report

The communication by Nowshin Farzana Khan et al., entitled “Evaluation of the safety and gastrointestinal migration of guandinylated chitosan after oral administration to ratspresents an interesting, valuable and well-conducted study on the in vivo behavior of the guanidinylated chitosan.

The study is fluently exposed in the manuscript, the materials and methods section is accurately disclosed and the obtained data are relevant and important.

My only concern is the GCS preparation method, which is extended to 7 days. Also, please mention what vigorously means in terms of rpm. Any preparation method that needs such a long period is hard to reproducible and apply in mass production. Are you sure that this is the only suitable time? Alternatively, do you take into consideration an improvement on the presented method?

Author Response

Reply to comments from reviewers:

We are appreciative of the reviewer’s helpful comments regarding our manuscript“Evaluation of the safety and gastrointestinal migration of guanidinylated chitosan after oral administration to rats” (Manuscript Number: jfb-2458289). We carefully considered their comments and prepared a revised version of our manuscript that takes these comments into account. The following is a point-by-point list of responses to the comments. In the revised manuscript, the revised sections or sentences are highlighted in red.

Reply to Reviewer #2:

We greatly appreciate the reviewer’s positive and constructive comments that were made.

  • In response to the comment, “My only concern is the GCS preparation method, which is extended to 7 days. Also, please mention what vigorously means in terms of rpm, we added the rotation speed for stirring (800 rpm) in the Materials and Methods.
  • Reply to the comment, “Any preparation method that needs such a long period is hard to reproducible and apply in mass production. Are you sure that this is the only suitable time?” Alternatively, do you take into consideration an improvement on the presented method?"

As described in our previous report (ref. 14), we investigated reaction conditions to obtain GCS with the highest degree of guanidinylation (DG). We should first note that we cannot use higher temperatures to prevent the guanidinylation between 1-amidinopyrazoles (reagent for guanidinylation). Therefore, this reaction required a long reaction period to obtain GCS. However, the reaction time (7 days) was longer than that for the end of the reaction to ensure that DG reached saturation. Therefore, the reaction conditions described in this manuscript (room temperature for 7 days) are suitable conditions for preparing GCS with ca. 50% DG (the highest DG) with good reproducibility. In addition, by using this method, we achieved a mass production of GCS (10 g<). These were described in our previous report (ref. 14-16).

Reviewer 3 Report

The reported work "Evaluation of the safety and gastrointestinal migration of guan- 2 idinylated chitosan after oral administration to rats" showed very intresting results which may help for future pre-clinical studies. All the related results are well organized. Thus it can be accepted in the present form. However, I couldn't see the zeta analysis datas/spectra in the whole paper. Pls consider it.

Author Response

Reply to comments from reviewers:

We are appreciative of the reviewer’s helpful comments regarding our manuscript“Evaluation of the safety and gastrointestinal migration of guanidinylated chitosan after oral administration to rats” (Manuscript Number: jfb-2458289). We carefully considered their comments and prepared a revised version of our manuscript that takes these comments into account. The following is a point-by-point list of responses to the comments. In the revised manuscript, the revised sections or sentences are highlighted in red.

Reply to Reviewer #3:

We much appreciate the reviewers for these positive and constructive comments.

  • In response to the comment, “The reported work "Evaluation of the safety and gastrointestinal migration of guan- 2 idinylated chitosan after oral administration to rats" showed very intresting results which may help for future pre-clinical studies. All the related results are well organized. Thus it can be accepted in the present form. However, I couldn't see the zeta analysis datas/spectra in the whole paper. Pls consider it.", we added Table1 with information on the zeta analysis results.

Table 1. Zeta potential of CS Lactate and GCS.

Formulations

Zeta potential (mv)

CS Lactate

13.29 ± 1.96

GCS

19.14 ± 0.86*

Values are expressed as mean ± SD (n = 4). P values were determined using one-way ANOVA. For all analyses, * p < 0.01 was considered to indicate statistical significance.

Regarding the quality of the English used in this paper: We had our paper examined by a U.S.-based editing service. A letter of confirmation attesting to this is attached.
